# The polyHIS Tract of Yeast AMPK Coordinates Carbon Metabolism with Iron Availability

**DOI:** 10.3390/ijms24021368

**Published:** 2023-01-10

**Authors:** Kobi J. Simpson-Lavy, Martin Kupiec

**Affiliations:** The Shmunis School of Biomedicine and Cancer Research, The George S. Wise Faculty of Life Sciences, Tel Aviv University, Tel Aviv 69978, Israel

**Keywords:** yeast, SNF1, iron metabolism, iron–sulfur clusters, pH, fermentation, respiration

## Abstract

Energy status in all eukaryotic cells is sensed by AMP-kinases. We have previously found that the poly-histidine tract at the N-terminus of *S. cerevisiae* AMPK (Snf1) inhibits its function in the presence of glucose via a pH-regulated mechanism. We show here that in the absence of glucose, the poly-histidine tract has a second function, linking together carbon and iron metabolism. Under conditions of iron deprivation, when different iron-intense cellular systems compete for this scarce resource, Snf1 is inhibited. The inhibition is via an interaction of the poly-histidine tract with the low-iron transcription factor Aft1. Aft1 inhibition of Snf1 occurs in the nucleus at the nuclear membrane, and only inhibits nuclear Snf1, without affecting cytosolic Snf1 activities. Thus, the temporal and spatial regulation of Snf1 activity enables a differential response to iron depending upon the type of carbon source. The linkage of nuclear Snf1 activity to iron sufficiency ensures that sufficient clusters are available to support respiratory enzymatic activity and tests mitochondrial competency prior to activation of nuclear Snf1.

## 1. Introduction

Iron is an essential element for life; however, Fe^2+^ is reactive and generates free radicals via the Fenton reaction. Therefore, iron levels and availability are tightly regulated [1]. In yeast cells, free Fe^2+^ is incorporated into the porphyrin ring of haem, whose functions include oxygen sensing [2], cytochrome function (both for lipid biosynthesis, e.g., ergosterol [3], and the electron-transport-chain of respiration [4]), as well as elimination of hydrogen peroxide by catalases [5]. Fe^2+^ is also utilized as iron–sulfur clusters, which are incorporated as co-factors in a variety of proteins. Iron–sulfur-containing proteins can be found in DNA metabolism (e.g.: DNA polymerases, the ribonucleotide reductase (RNR) subunit Rnr2 (the RNR catalyzes the limiting step in dNTP synthesis)) [6], carbon metabolism (e.g., Aco1, Sdh4) [7], and the ribosome cycle [8]. Thus, under conditions of iron scarcity, cells must prioritize which class of protein receives iron.

The bioavailability of iron is tightly regulated. Under iron-replete conditions, the Yap5 transcription factor (TF) induces expression of genes, such as those encoding the vacuolar iron-importer Ccc1, to sequester excess iron [9,10]. Ccc1 imports Fe^2+^ into the vacuole, where it is oxidized and stored as Fe^3+^ [11]. Upon iron starvation, iron is restored from the vacuole back to the cytoplasm [12].

Under low-iron conditions, yeasts express a set of genes known as the (low) iron regulon, which govern iron uptake and homeostasis. The expression of these genes is controlled by the Aft1 and Aft2 transcription factors (TFs). Aft1 and Aft2 show partial redundancy, but Aft1 is more involved in the expression of iron transporters (such as *FET3*), whereas Aft2 regulates the expression of genes, such as *SMF3* (vacuole exporter) and *MRS4* (mitochondrial importer) which are involved in intracellular iron distribution [13]. Aft1 also upregulates expression of the ribonucleotide reductase subunit *RNR1* [14] and *CTH1/2*, which degrades select mRNAs that are mainly associated with carbon respiration and amino acid biosynthesis [7,15]. Thus, the low-iron response regulates the apportioning of iron to different processes.

Iron–sulfur complexes are exported from the mitochondria, and under conditions of sufficient iron–sulfur cluster provision are recognized by the glutaredoxins Grx3 and Grx4 (either one; deletion of both is lethal), which form a sandwich, comprising two Grx3/4 moieties, both binding a single iron–sulfur cluster. Following replacement of one of the glutaredoxins by Fra1 or Fra2 (Bol2), this complex binds to Aft1 or Aft2, inhibiting the DNA binding ability of these TFs [16,17,18,19]. Thus, under conditions where iron–sulfur clusters are available, the iron regulon is inactive.

Although Aft1 and Aft2 share a core, Fe–S cluster-regulated DNA binding domain, Aft1 has extended sequences at both the N and C termini that are absent from Aft2 [20]. In addition to activating the iron regulon, Aft1 plays central roles in seemingly unrelated processes: it interacts with the kinetochore protein Iml3 to promote peri-centromeric cohesion [21], with Sit1 (Arn3) to regulate ferroxamine B uptake [22], and with the DNA damage checkpoint protein Rad9 to monitor fragile genomic sites [23].

The yeast Snf1 kinase (AMPK in mammals) promotes the switch from fermentation of hexoses (e.g., glucose or fructose) to produce ethanol, to the respiration of poor carbon sources (such as glycerol, lactate, ethanol, etc.) [24]. Genes involved in gluconeogenesis or in the respiration of alternative carbon are regulated by Snf1 via the activation of transcription factors. Examples of upregulated factors are Adr1, which regulates genes such as *ADH2* (utilized in this paper as a reporter for Snf1 activity) [25] and Cat8 (to express, e.g., *FBP1*, *PCK1*) [26]. Snf1 also inhibits repressors, such as Mig1 and Nrg1 (to derepress *SUC2*, *GAL1* [27,28]). In addition, Snf1 inhibits Acetyl-CoA carboxylase (Acc1) to downregulate fatty acid synthesis [29], Psk1 to direct glucose-6-phosphate to cell wall construction [30], and inhibits adenylate cyclase (Cyr1) [31]. Snf1 also phosphorylates and inhibits the arrestins Rod1 and Rog3 to regulate the stability of plasma membrane carbon transporters [32].

Snf1 is activated by phosphorylation at T210 upon glucose deprivation [33]; this site becomes dephosphorylated when glucose is available [34]. When glucose is absent, a gamma-protein activator (Snf4) associates with Snf1, and the localization of the SNF1/AMPK complex within the cell (to the vacuolar membrane, plasma membrane and nucleus, respectively) is determined by one of three beta-localizing proteins: Sip1, Sip2, or Gal83 [35]. Localization of Snf1 to the nucleus is inhibited by glucose [36]. Glucose also contributes to inhibition of Snf1 via PKA activation of Mms21 which SUMOylates Snf1at lysine 549 [37].

The Snf1 protein comprises a kinase domain (aa 54–391) (KD) and a regulatory domain (aa 392–633) (RD) [38] (Appendix A). In the presence of glucose, the kinase domain (KD) is inhibited by interaction with the regulatory domain (RD). Upon phosphorylation of Snf1 at T210, Snf4 interacts with aa 460–498 of Snf1 (RD-γ), preventing the interaction of the kinase and regulatory domain, and disinhibiting Snf1 kinase activity [38,39]. The beta proteins (Sip1, Sip2, Gal83) interact with aa 515–633 of Snf1(RD-β) [40]. Snf1 associates with Snf4 and one of the beta proteins to produce a stable, active SNF1/AMPK complex [35]. The pre-kinase region (PKR) comprises the amino acids from the N-terminus until the kinase domain. In *S. cerevisiae*, the PKR comprises amino acids 1–53 and contains a polyhistidine tract comprising 13 contiguous histidines (followed by glycine and one more histidine) [41] that interact with RD-β in response to deprotonation caused by the glucose activation of Pma1. This functions as a progressive pH-sensing module (PSM) that controls Snf1 activity in response to glucose deprivation [42]. In this work, we show that the protonated form of the PSM also integrates iron deprivation signals to inhibit Snf1 activity.

## 2. Results

### 2.1. The PSM Receives a Signal Transmitted by Iron Deprivation

Histidine has been reported to form ligands with transition metal ions, such as Ni^2+^ and, in combination with cysteines, Fe^2+^ [43]. Although we have already shown that the polyhistidine tract in the PSM of Snf1 is a pH sensor [42], we wondered whether it might also have a role in metal abundance sensing. We found that EDTA, a chelator mainly of Mg^2+^ and Ca^2+^, did not affect *ADH2* expression in a polyHIS-dependent manner (Appendix A). However, addition of the iron chelator BPS (0.1 mM) lowered *ADH2* expression by approximately 50% whilst the addition of Mohr’s salt (ferrous ammonium sulfate, 5 mM) increased *ADH2* expression (Figure 1A). The effects of iron supplementation or chelation upon *ADH2* expression are dependent upon the polyHIS tract of Snf1, since deletion of these amino acids (ΔH), or their replacement by alanine (A), increased *ADH2* expression and uncoupled *ADH2* expression from iron availability (Figure 1A). Iron supplementation did not rescue *ADH2* expression when the polyHIS tract was substituted with aromatic amino acids that have previously been shown to be inactive [42], such as the Snf1^W^ or Snf1^Y^ mutant proteins (Figure 1A). In our previous work, we found that decreasing the number of histidines in the PSM progressively increases Snf1 activity by decreasing the interaction between the NTD and the CTD [42]. We therefore measured the response to BPS in Snf1^4H^ and Snf1^8H^ cells (PSM with only four or eight histidines) and found that the inhibition of Snf1 (as determined by *ADH2* expression) by iron deprivation increased when more histidines were present (Figure 1B). It is possible that iron deprivation could exert its effects by diminishing phosphorylation of threonine 210 at the active site, which is also needed for Snf1 activity. However, iron chelation with BPS did not affect T210 phosphorylation, neither in WT cells nor in cells expressing Snf1^ΔH^ (Figure 1C and Appendix A). Although Snf1^ΔH^ was much less abundant than Snf1^WT^ (Figure 1C and Appendix A), it is three times more phosphorylated at T210 (Figure 1C and Appendix A)—Snf1 abundance was not affected by iron deprivation (Figure 1C and Appendix A). In response to lack of glucose, Snf1 activity results in the activation of Adr1 by dephosphorylation at S230 [25]. Indeed, Adr1^S230A^ cells, unable to undergo phosphorylation, do not require Snf1 for *ADH2* expression [25]. Expression of Adr1^S230A^ resulted in the suppression of the lowered *ADH2* expression due to BPS treatment (Figure 1D), suggesting that iron deficiency is sensed upstream of Adr1.

Since the polyHIS tract within the pre-kinase region is crucial for mediating the inhibition of Snf1 in response to iron–sulfur cluster deficiency, we considered whether overexpressing this region alone might affect *ADH2* expression. The PKR was overexpressed in a 2μ plasmid from an *ADH1* promoter; we created a nuclear version of this region by fusing it to the Gal4 binding domain (GBD), or left it intact, thus directing it to the cytoplasm. An overexpression of nuclear Snf1^PKR^ increased *ADH2* expression and rendered *ADH2* expression insensitive to iron chelation (Figure 1E). This was dependent upon the polyHIS tract, since an overexpression of Snf1^PKRΔH^ or Snf1^PKR-A^ did not affect *ADH2* expression. Interestingly, an overexpression of cytoplasmic Snf1^PKR^ also did not affect *ADH2* expression (Figure 1E). Therefore, it seems that the overexpression of nuclear Snf1^PKR^ saturates a binding site in an inhibitor protein via the polyHIS tract, and that this inhibitor protein is nuclear. Since overexpression of the positively charged Snf1^PKR-R^ allele, where the polyHIStidines are replaced by poly-arginine, also increased *ADH2* expression (Figure 1E), the binding site of the inhibitory protein with Snf1^PKR^ is likely to be negatively charged.

### 2.2. The Transcription Factor Aft1 Transmits a Low-Iron Signal to Snf1

Under conditions of sufficient iron–sulfur cluster production, iron–sulfur clusters are bound by two Grx3 or Grx4 proteins, and then one is swapped for Bol2. The Grx3/4-FeS cluster-Bol2 sandwich serves as an inhibitor of the Aft1 and Aft2 TFs [16,18,19,20,43] by causing dissociation from DNA [44], phosphorylation, and Msn5-mediated export from the nucleus [45]. We therefore examined the effects of deletion of the Aft TFs on *ADH2* expression. The iron-sensing and DNA-binding parts of Aft2 are also present in Aft1, but Aft1 also possesses extended acidic, basic, and polyQ regions (Appendix A). Deletion of *AFT1,* but not its paralogue *AFT2,* increased *ADH2* expression in WT cells to the level seen in *Snf1^ΔH^* or *Snf1*^A^ cells, whereas expression of the hyperactive and constitutively nuclear Aft1^1up^ allele as the sole copy of Aft1 diminished *ADH2* expression but was suppressed by *Snf1^ΔH^* or *Snf1*^A^ (Figure 2A). Deletion of *AFT1* did not affect the interaction of Snf1 and Gal83 for Snf1^WT^, nor for hyperactive Snf1^ΔH^ or Snf1^A^, and nor for Snf1^W^ or Snf1^Y^ which have attenuated activity and interaction (Figure 2B). This suggests that Aft1 (but not Aft2) transmits the low-iron signal to regulate of Snf1 activity (rather than direct iron–polyHIS interaction) but does not regulate the interaction between Snf1 and the β-subunits of the AMPK complex.

Ethanol is oxidized by Adh2 to acetylaldehyde, which is then oxidized by Ald2-6 to produce acetate (and either NADH or NADPH), which then is conjugated to coenzyme A (producing cytoplasmic acetyl-CoA) by Acs1 and Acs2. Even when cells are respiring ethanol, more acetate is produced than is consumed by conversion to acetyl-CoA [46]. Acetate is toxic and directly activates the Haa1 transcription factor [47] to induce expression of genes, such as *YRO2* [48]. Haa1 localization has been used as a biosensor for external acetate [49] and we utilized the expression of *YRO2* as a biosensor for both external and internal acetate [46] and thus also for ethanol catabolism [46]. As reported previously, wild-type cells produced and accumulated acetate (as measured by *YRO2* expression) from both glucose and ethanol catabolism (both metabolic pathways produce acetylaldehyde), but not from oleate catabolism, which directly produces acetyl-CoA [46] (Figure 2C). Iron deprivation, as expected, clearly lowered *YRO2* expression in wild-type cells. However, this was not observed in Snf1^ΔH^ or *aft1Δ* cells (Figure 2C). This demonstrates that the lowering of the rate of ethanol catabolism by iron deprivation is regulated by Aft1 and the polyHIS tract of Snf1.

Since Aft1, but not Aft2, regulates Snf1 activity, we compared the protein sequences for different features (Appendix A). We considered that the polyHIS tract, which is protonated in the absence of glucose [42], might interact with a cluster of negative polar or acidic residues. This hypothesis is supported by the increased *ADH2* expression when Snf1^PKR-R^ is overexpressed (Figure 1E), which could chelate an inhibitory negatively charged protein. Such a region is found between aa 16 and 24 (inclusive) of Aft1 (deleted in Aft1^Δ9^), with a more extended region stretching from aa 16 to 36 (inclusive) (deleted in Aft1^Δ24^). Both Aft1^Δ9^ and Aft1^Δ24^ are more active than wild-type Aft1, as determined by *FET3* expression upon iron chelation by BPS (Appendix A). This experiment was conducted in glucose medium since glucose starvation inhibits *FET3* expression (Appendix A). Overexpression of Aft1 decreased Snf1 activity (in a polyHIS-dependent manner), whereas deletion of the 9aa serine/threonine/acid stretch of Aft1 (*AFT1^Δ9^*) not only suppressed the overexpression of Aft1 but further increased Snf1 activity to that seen in *snf1^ΔH^* or *aft1*Δ cells (Figure 2D). Deletion of aa16–36 of Aft1 (*AFT1^Δ21^*) did not further increase Snf1 activity.

Yeast two-hybrid assay (whereby both proteins are constitutively nuclear) showed an ethanol-dependent interaction between Snf1 and Aft1, which was lowered (but not abolished) in Aft1^Δ9^ cells (Figure 3A). Aft1^1–44^ was sufficient for interaction with Snf1, and this was diminished by deletion of amino-acids 16–24 (Aft1^1–44Δ9^) (Figure 3B). Therefore, this negatively polar region of Aft1 is necessary and sufficient for interaction with Snf1. Elevated iron concentration or iron deprivation did not affect these interactions, suggesting that once Aft1 has entered the nucleus, iron no longer regulates the interaction of Aft1 and Snf1. Thus, the role of iron is to regulate the localization of Aft1. In contrast, the interaction of nuclear Aft1 and nuclear Snf1 is still regulated by carbon source.

Next, we determined which parts of Snf1 regulate the interaction with Aft1. The full-length Snf1 protein interacted with Aft1 in an ethanol-containing medium. The interaction required the polyHIS tract, since it was abolished by its deletion, and also required T210 phosphorylation (Figure 3C). The interaction of full-length Snf1 in glucose was about 15% of that in ethanol. Substitution of the 14 histidines for non-protonatable tryptophan likewise impeded Snf1/Aft1 interaction (Figure 3C).

Aft1 did not interact with the regulatory (CTD) domain of Snf1. Truncation of Snf1 to leave only the N-terminal domain (NTD) elevated the interaction levels in glucose and even more so in ethanol media. This was probably due to a loss of the competing β-subunit binding site in the regulatory domain of Snf1. This interaction was polyHIS-dependent, and substitution of the 14 histidines for non-protonatable tryptophan still impeded the Snf1^NTD^/Aft1 interaction. Interestingly, this interaction did not require T210 phosphorylation, indicating that once Snf1 is in the open conformation under respiratory conditions, T210 phosphorylation is not required (Figure 3C). The PKR alone was sufficient to interact with Aft1. This interaction was also elevated in ethanol media and required the polyHIS. Substitution of the 14 histidines for non-protonatable tryptophan still impeded the Snf1^NTD^/Aft1 interaction. Ethanol still resulted in greater Snf1/Aft1 interaction than glucose media, even when the CTD was absent, or when just amino acids 1–53 (the PKR) were present, and the interaction between Snf1 and Aft1 did not occur when the histidines were substituted with tryptophan (Figure 3C); protonation of the polyHIS to give it a positive charge is involved in the interaction of Snf1 with Aft1.

To determine whether intracellular pH regulation by Pma1 is involved in the interaction of Snf1 with Aft1, we hyper-activated Pma1 by deleting *HSP30* and by truncating Pma1 (Pma1-Δ901) [42] to mimic the glucose state (which deprotonates the polyHIS tract). Hyper-activation of Pma1 lowered the Snf1^FL^/Aft1 interaction (Figure 3D). Since the polyHIS of Snf1^FL^ interacts with the β-subunit binding site in the regulatory domain of Snf1 [42], we also used Snf1^NTD^, which does not have a β-subunit binding site to compete with Aft1 for polyHIS interaction. Hyper-activation of Pma1 dramatically lowered the Snf1^NTD^/Aft1 interaction (Figure 3D). Thus, it seems that not only does polyHIS protonation release the polyHIS tract from the β-subunit binding site in the regulatory domain of Snf1 [42], but the same protonation also enables the interaction of the polyHIS tract with Aft1.

### 2.3. Nuclear Aft1 Inhibits Snf1

We have established that iron deprivation inhibits Snf1 (in a polyHIS-tract-dependent manner) and that the agent is the Aft1 transcription factor. The Hog1 kinase phosphorylates Aft1 at S210 and S224 to export Aft1 from the nucleus. This activity does not seem to be linked to the canonical Hog1 response to hyperosmolarity [50]. These sites were previously demonstrated to be dephosphorylated upon iron depletion, causing Aft1 relocalization into the nucleus [45]. We used a phospho-mimicking aspartate Aft1 mutant (Aft1^SS210,224DD^ (hereafter referred to as Aft1^DD^) [45]) to determine the effects of excluding Aft1 from the nucleus upon *ADH2* expression. Aft1^DD^ increased *ADH2* expression (Figure 4A) and is not additive with Snf1^ΔH^. Aft1^DD^ also uncoupled *ADH2* expression from iron concentrations (Figure 4A). Conversely, the nucleus-retained phospho-null Aft1^S210A,S224A^ mutant (Aft1^AA^) decreased *ADH2* expression. *ADH2* expression in cells expressing *aft1*^AA^ was not further diminished by BPS (Figure 4A). This suggests that it is nuclear Aft1 that inhibits Snf1, as opposed to the iron deficiency per se. As expected, the nuclear-retained Aft1^AA^ shows increased *FET3* expression, and the nuclear-excluded Aft1^DD^ shows decreased *FET3* expression (Appendix A).

When sufficient iron is available, Aft1 dissociates from DNA [44] and undergoes phosphorylation and Msn5-mediated export from the nucleus [45]. We deleted *MSN5* to determine whether phosphorylation of Aft1 suffices to prevent inhibition of Snf1, or whether the export of Aft1 from the nucleus is required. Deletion of *MSN5* suppressed the increased *ADH2* expression in Snf1^WT^ cells but not Snf1^ΔH^ cells (Figure 4B), showing that nuclear export of Aft1, rather than phosphorylation, lowers the inhibition of Aft1. This suggests that nuclear Aft1 inhibits Snf1.

Since both Snf1 and Aft1 are present throughout the cytoplasm (and the nucleus under appropriate conditions), we used a PCA Venus assay [51] to determine whether these proteins interact, under which conditions, and where this interaction occurs (Figure 4C). In this assay, interacting proteins tagged with two halves of a Venus fluorescent protein (VF1 and VF2) reconstitute an active Venus protein in an irreversible fashion, provided they are close enough [51]. *snf1Δ aft1Δ* cells expressing VF1-Snf1 and Aft1-VF2 were grown overnight in either glucose or glycerol with either 5 mM Mohr’s salt or 100 μM BPS and diluted the following morning in the same conditions. On the first day, glucose-grown cells did not show any interaction, whereas glycerol-grown cells showed a distinct interaction that colocalized with the nuclear pore marker Nup49-Cherry. This interaction occurred both in iron-rich and iron-limited media. On the second day, by which time glucose was exhausted, BPS-grown cells showed several foci of interaction (not at the nuclear membrane) but this did not occur in the presence of Mohr’s salt (Figure 4C and Appendix A). In contrast, VF1-Snf1^ΔH^ did not interact with Aft1-VF2 at the nuclear membrane in glycerol-grown cells; rather, under these conditions, cells had some cytoplasmic puncta similar to wild-type VF1-Snf1 interacting with Aft1-VF2 in glucose + BPS, and no interaction at all in glucose. Likewise, forcing Aft1-VF2 into the cytoplasm by using the Aft1^DD^-VF2 mutant also prevented interaction at the nuclear membrane with VF1-Snf1 (Figure 4C). Thus, the interaction between Aft1 and Snf1 is restricted to the nuclear membrane despite the proteins being both either cytoplasmic or nuclear (depending upon the conditions) and requires the polyHIS motif of the pre-kinase region of Snf1. Moreover, Aft1 must enter the nucleus (implying that the interaction occurs on the nucleoplasm side of the nuclear membrane), and the interaction is regulated by carbon source. This is somewhat reminiscent of Mig1 repressor activity requiring interaction with the nuclear pore complex components Nup120 and Nup133 [52]. Indeed, deletion of either Nup120 or Nup133 abolished the Snf1^PKR^/Aft1 interaction (we used Snf1^NTD^ to prevent interference from the regulatory domain) (Figure 4D) and suppressed the reduction in *ADH2* expression caused by iron deprivation (Figure 4E) showing that the Aft1/Snf1 interaction and inhibition of Snf1 requires intact nuclear pores.

### 2.4. Spatiotemporal Regulation of Snf1 by Iron

Some transcription factors regulated by Snf1 are nuclear (such as Adr1, irrespective of glucose status) [53] while others shuttle between the nucleus and cytoplasm, with nuclear distribution increasing in the absence of glucose (such as Mig1) [54]. The former require nuclear Snf1 for their regulation, whereas the latter can be regulated by Snf1 irrespective of the location of Snf1.

The beta subunits of the SNF1 complex regulate its localization [36], with Sip1 directing Snf1 to the vacuolar membrane, Sip2 to the plasma membrane, and Gal83 to the nucleus. We used this to control the localization of active Snf1, creating *gal83Δ* and *sip1Δ sip2Δ* strains whose active SNF1 is either excluded from the nucleus or nucleus-enriched [36], respectively. *ADH2* expression in the absence of Gal83 was about one-quarter that of wild-type cells (note that this is sufficient for growth on poor carbon sources), and deletion of *GAL83* suppressed the increased *ADH2* expression of *snf1^ΔH^* cells, showing that nuclear Snf1 is indeed required for *ADH2* expression. However, the remaining 25% of *ADH2* expression in *gal83Δ* cells was independent of iron status (Figure 5A), suggesting that non-nuclear Snf1 is not iron-regulated. *ADH2* expression was surprisingly decreased in *sip1Δsip2Δ* cells; the low *ADH2* expression in these cells was still further decreased by iron deprivation and this was suppressed by Snf1^ΔH^ (Figure 5A).

Expression of genes required for the metabolism of mono- and di-hexoses other than glucose or fructose, such as maltose (*MAL*), sucrose (*SUC*), and galactose (*GAL*), are repressed by Mig1 under glucose conditions. Indeed, *SUC2* (invertase) can be considered the classic Snf1 and Mig1 target gene [52,55]. *SUC2* expression was markedly higher in ethanol media compared with sucrose, but was not affected by deletion of *GAL83*, nor was it increased in Snf1^ΔH^. Depletion of iron did not affect *SUC2* expression (Figure 5B). Indeed, when active Snf1 was restricted to the nucleus (*sip1Δsip2Δ* cells), *SUC2* expression was increased and became iron- and polyHIS-regulated (Figure 5B,C). *SUC2* expression in ethanol media was twice the level of the *SUC2* expression in sucrose; however, the same expression pattern was observed. Thus, the enrichment of active Snf1 into the nucleus renders Mig1-regulated genes iron-regulated.

### 2.5. Non-Nuclear Snf1 Substrates Are Not Regulated by Iron Deprivation

In addition to regulating gene expression, cytoplasmic Snf1 also regulates metabolic enzymes (such as inhibiting Acc1—the first step of fatty acid synthesis [56]) and the stability of carbon transporters (such as inhibiting Jen1 (lactate/acetate importer) degradation in the vacuole [32,57]).

Cytoplasmic Snf1 phosphorylates the arrestin Rod1, preventing it from playing its role in the ubiquitylation of plasma membrane proteins, such as Jen1 by Rsp5 and their subsequent endocytosis and degradation in the vacuole [57]. Iron deprivation did not affect the phosphorylation of Rod1 upon transfer of cells from the glucose media to the ethanol media (Figure 6A), nor did the hyperactive alleles of Aft1—Aft1^Δ9^ (Figure 6A), Aft1^AA^ (Figure 6B), the less-active Aft1^DD^ (Figure 6C), or even the deletion of *AFT1* (Figure 6D). Snf1^ΔH^ did not affect the phosphorylation of Rod1 upon glucose deprivation either alone or in combination with the Aft1 mutants (Figure 6A–E).

Aft1 mutants that increase (Aft1^AA^) (Figure 6B) or decrease (Aft1^DD^) (Figure 6C) Aft1 activity by changing Aft1 localization did not affect Snf1 abundance. However, the Aft1^Δ9^ mutant (Figure 6A,E) which has reduced interaction with Snf1 (Figure 3A), or absence of Aft1 (Figure 6D), resulted in a decrease in Snf1 abundance. Similarly, Snf1^ΔH^ was also expressed at lower levels than Snf1^WT^. Combining Snf1^ΔH^ with Aft1 mutations did not further increase Snf1 instability (Figure 6B–E). This suggests that the interaction between Snf1 and Aft1 stabilizes Snf1.

We monitored the localization of Jen1-GFP (expressed from a *GAL1* promoter). Whilst glucose addition caused the disappearance of Jen1-GFP from the plasma membrane, iron depletion did not affect Jen1-GFP localization (Figure 6F and Appendix A). As expected, increasing Snf1 activity by deletion of *REG1* or the deletion of the arrestin *ROD1* stabilized Jen1 even in the presence of glucose (Figure 6F). Together, these results suggest that iron–sulfur cluster depletion only inhibits nuclear Snf1 activities while cytoplasmic Snf1 is not regulated by iron.

## 3. Discussion

We have discovered a mechanism by which carbon and iron metabolism are jointly coordinated. Under respiration conditions, the polyHIS tract in the progressive pH-sensing module (PSM) at the N-terminus of Snf1 is protonated and interacts with the N-terminus of nuclear Aft1.

Under conditions of iron–sulfur cluster limitation, Snf1-regulated genes involved in respiration can be considered as a competitor for scarce iron–sulfur clusters with other enzymatic processes (such as DNA replication and repair). It has previously been reported that low iron correlates with a downregulation of respiratory gene expression. Furthermore, reducing iron–sulfur cluster production by mutation of *ISU1* increases the fermentation of xylose [58,59], and *isu1* mutation shifts cells from respiration to fermentation [60]. The results in this paper provide evidence for the mechanism by which low iron causes the downregulation of respiration gene expression and ethanol catabolism via the interaction of Aft1 with the polyHIS tract of Snf1 (Figure 7).

### 3.1. The polyHIS Tract Has Different Interactions Depending upon Carbon Source and pH

We have previously identified an internal interaction of the deprotonated polyHIS tract with the β-subunit binding site in the regulatory domain of Snf1. This interaction inhibits Snf1 in response to cytoplasmic alkalization due to Pma1 activity in the presence of glucose [42]. We have now demonstrated that the protonated polyHIS tract (caused by cytoplasmic acidification following Pma1 downregulation due to glucose withdrawal) interacts with Aft1 (Figure 3C,D). The interaction between Snf1 and Aft1 requires the polyHIS tract to be freed from interaction with the regulatory domain, either by phosphorylation at T210 or by truncation of the regulatory domain (Figure 3C). However, the interaction between polyHIS and Aft1 can still be inhibited by Pma1 hyper-activation (Figure 3D). This multitasking of the polyHIS tract allows Snf1 to integrate diverse metabolic signals to effect an appropriate response to suboptimal environmental conditions

### 3.2. The polyHIS Tract Links Mitochondrial Status to Nuclear Gene Expression

Iron–sulfur cluster insufficiency results in Aft1 nuclear localization, and this same condition also lowers nuclear Snf1 activity (Figure 1A,B,D,E). Furthermore, cells expressing the constitutively active Aft1^up^ allele fail to grow on respiratory media (glycerol) [61]. Under this dual condition of growth on poor carbon sources and iron deprivation, a nine amino acid negative polar motif of Aft1 interacts with the exposed polyHIS tract of Snf1 and thereby inhibits Snf1 activity (Figure 2D–F), thus linking the mitochondrial process of iron–sulfur cluster production to Snf1 activity. This interaction of Aft1 and Snf1 can only occur with active Snf1 in the nucleus since contact of the regulatory domain (aa 392-633) of Snf1 with the PSM (as occurs in inactive Snf1) downregulates the interaction with Aft1 (Figure 3C). Phosphorylation at T210 serves to activate and disengage the PSM from RD-β. If Snf1 is truncated so it lacks the RD (so only Snf1^NTD^ is present), then phosphorylation at T210 is no longer required for the interaction of Aft1 with Snf1 (Figure 3C). Aft1 and RD-β are competitors for interaction with the PSM—but these interactions are also regulated by pH. Hence, unprotonated PSM (e.g., in high Pma1 activity, glucose) and Snf1^W^ interact with RD-β [42], but protonated PSM (low Pma1 activity, ethanol) but not Snf1^W^ interacts with Aft1 (Figure 3C,D). Thus, a single motif (polyHIS) engages in different interactions to integrate carbon and iron status.

This linkage of nuclear Snf1 activity to iron–sulfur cluster sufficiency serves at least two purposes: (1) to ensure that sufficient iron–sulfur clusters are available to support respiratory enzymatic activity and (2) to determine mitochondrial competency prior to activation of nuclear Snf1 by having the iron signal be a product of a mitochondrial process.

### 3.3. Restricted Localization of the Snf1-Aft1 Interaction

How does iron deprivation cause inhibition of Snf1? It is possible that free Aft1 is an inhibitor of Snf1, regardless of its localization. We forced Aft1 to be nuclear via three different mechanisms: Aft1^AA^ (Figure 4A), deletion of *MSN5* (Figure 4B), or GBD-Aft1 (Figure 1E)]. In all three cases the interaction with, and inhibition of, Snf1, occurs independently of the iron availability. Furthermore, Aft1^DD^ still inhibited Snf1 in *msn5Δ* cells (Figure 4B), showing that it is the export of Aft1 from the nucleus, as opposed to its phosphorylation, that prevents Aft1 from inhibiting Snf1.

The nuclear pore complex is increasingly seen as an organizing center for protein–protein and protein–DNA interactions. For example, double-strand break repair takes place at the inner nuclear membrane side of the nuclear pore [62,63]. The repair of sub-telomeric double-strand breaks is also dependent on both Nup120 and Nup133 by anchoring telomeres to the nuclear membrane [64]. The interaction of Mig1 with its targets also requires Nup120 and Nup133 [52]. Although (in respiration with low-iron conditions) both Aft1 and Snf1 are found throughout the cell, the interaction between them occurs at the nuclear membrane (Figure 4C), requiring Nup120 and Nup133 (Figure 4D), and the absence of *NUP133* uncouples the regulation of *ADH2* expression by iron (Figure 4E). The interaction between Aft1 and Snf1 occurs at the nuclear membrane and is regulated by carbon source, but not by iron availability (Figure 4C). However, nuclear import of Aft1 is required for the inhibition of Snf1 activity.

### 3.4. Spatial Regulation of Snf1 by Iron

Although *ADH2* expression is inhibited by iron deprivation in a PSM-dependent manner (Figure 1A) and requires Gal83 (nuclear Snf1) (Figure 5A), *SUC2* expression is iron-independent and polyHIS-tract-independent (Figure 5B,C). This also confirms prior results [41] using invertase (Suc2) activity as a reporter for Snf1, which stated that deletion of the polyHIS tracts does not decrease invertase activity. *SUC2* expression is also Gal83-independent (Figure 5B,C).

The remaining 25% of *ADH2* expression observed in *gal83* cells is also polyHIS- and iron-independent (Figure 5A); therefore, exclusion of Snf1 from the nucleus prevents low-iron-induced inhibition of Snf1. In contrast, *SUC2* expression in the cytoplasm is normally neither iron- nor polyHIS-regulated, but upon deletion of both *SIP1* and *SIP2, SUC2* expression is elevated and becomes both low-iron- and polyHIS-regulated (Figure 5B,C).

While Adr1 and Cat8, which regulate *ADH2* expression, are nuclear, Mig1, which regulates *SUC2* expression, is both nuclear and cytoplasmic. Indeed, Gal83 remains cytoplasmic during the growth of cells on sucrose (or galactose) [36]. In the discussion from Vincent et al. 2001 [36], there is speculation as to whether differential localization of Snf1 might allow a difference in response under different environmental conditions. Our results show that only nuclear Snf1 is inhibited by iron deprivation via Aft1, whereas Mig1-regulated genes are iron-independent. Other cytoplasmic roles of Snf1, such as the inhibition of Rod1 (Figure 6A–E) and thus retention of the Rod1 target Jen1 at the plasma membrane (Figure 6F), are also unaffected by iron deprivation. This temporal–spatial mechanism of Snf1 inhibition by iron deprivation permits the Snf1-dependent fermentation of carbon sources, such as galactose, sucrose, and maltose, even under low-iron conditions, while inhibiting the iron-intense respiration of poor carbon sources, such as glycerol, lactate, or ethanol.

### 3.5. Atf1 Regulates Snf1 Stability

Snf1 is unstable in the absence of Ubp8 in the presence of cycloheximide due to ubiquitylation of SUMOylated Snf1 [65]. Here, we observe that Snf1^ΔH^ is found at 25% of the levels of Snf1^WT^ (Figure 1C, Figure 6A,E, and Appendix A) and yet is three times more phosphorylated at T210 (Figure 1C, Figure 6A,E, and Appendix A). This instability is due to a lack of interaction with Aft1, since Snf1 abundance is lowered in *aft1Δ* or Aft1^Δ9^ (in ethanol) cells but not Aft1^AA^ or Aft1^DD^ cells. Deletion of both the polyHIS tract and the 9 amino acid Snf1-interacting motif of Aft1 does not further destabilize Snf1. Indeed, the transfer of Aft1^Δ9^ cells from glucose to ethanol media rapidly results in Snf1 destabilization (within 30 min) (Figure 6A). This instability of Snf1 is iron-independent. Thus, Aft1 regulates Snf1 protein levels (iron independent) and activity (iron dependent). Excessive Snf1 activity is toxic—the lethality of deletion of two phosphatase components that inactivate Snf1 (Reg1 and Sit4) is rescued by a further deletion of *SNF1* [66]. Our results suggest that Aft1 is also involved in the control of Snf1 toxicity, and this mechanism provides a safety brake whereby excessive Snf1 activity (toxicity) is countered by decreased Snf1 stability.

### 3.6. Iron and the Diauxic Shift

Glucose exhaustion (diauxy) has been reported to activate Aft1 to induce expression of five iron-regulon genes (*FET3*, *FTR1*, *TIS11*, *SIT1*, and *FIT2)* [67]. This requires Snf1, but SNF1 is not responsible for Aft1 phosphorylation under these conditions [67,68]. Although growing cells ab initio in glycerol results in an interaction of Snf1 and Aft1 and the inhibition of Aft1, this interaction does not occur under conditions of glucose exhaustion (Figure 4C and Appendix A). This may be due to other metabolites or metabolic signaling taking place during glucose exhaustion. Indeed, glycerol-grown cells failed to express *FET3* following iron depletion (Appendix A). Furthermore, iron sufficiency has also been reported for passage through the diauxic shift [69] and inhibition of iron-cluster production promotes fermentation over respiration [58,59,60]. Iron deficiency also impairs mitochondrial function through Cth2, which targets specific mRNAs involved in respiration for degradation [61]. Iron deficiency lowers the rate of conversion of ethanol to acetate (Figure 2C). Thus, this work demonstrates that iron and Snf1 signaling are tightly coordinated to reduce futile adverse communication.

### 3.7. Comparison with Other Species

The pathogenic yeast *Candida glabrata* possess a low-iron Aft1/2 transcription response system similar to that of *S. cerevisiae*, albeit with additional pathways also found in other *Candida* species [70]. Since *Cg*Snf1 also contains a polyHIS tract in its PKR of 12 contiguous histidines [42], and *Cg*Aft1 has an acidic region at aa61–67 (EELTEEE), a similar regulation of *Cg*Snf1 by *CgAft1* would be expected. More broadly, iron and carbon metabolism are linked throughout all kingdoms, with iron deprivation leading to an inhibition of the Krebs cycle enzyme synthesis by the sRNA RyhB in *E. coli*, forcing cells to ferment rather than respire [71,72]. Similarly, iron deprivation leads to a shutdown of the Krebs cycle in *Mycobacterium tuberculosis* [73].

Iron overload results in reactive oxygen species generation due to the Fenton reaction. Iron in the bound form of ferritin is elevated in the brains of Huntington’s Chorea patients, especially in the basal ganglia, where it is associated with disease progression [74]. Excessive iron has been mechanistically implicated in the deleterious biochemistry of neurons in Parkinson’s disease, where it not only generates ROS but also induces α-synuclein accumulation [75] and dopamine oxidation [76]. Iron overload is also a driving cause of metabolic syndrome and Type II Diabetes [77]. Mice fed a high-iron diet exhibit increased AMPK T172 phosphorylation (equivalent to T210 in *S. cerevisiae* Snf1) and increased phosphorylation of the Snf1 target Acc1 both in skeletal muscle and liver [78]. Iron overload activates the LKB1 kinase (which phosphorylates Snf1 at T172) by increasing SIRT1 activity, resulting in de-acetylation of LKB1 [78]. Thus, lowered protein acetylation (indicative of low acetyl-CoA levels) stimulates AMPK to increase acetyl-CoA production, and iron overload locks this feedback loop as active. Elevated activation of AMPK by excess iron (determined both by T172 phosphorylation and Acc1 phosphorylation) has also been reported in mesenchymal stromal cells and is implicated in Myelodysplastic Syndrome. High, dysregulated AMPK activity in these cells results in mitochondrial fragmentation and apoptosis [79]. On the other hand, iron deprivation inhibits isoproterenol-induced lipolysis in 3T3-L1 adipocytes [80]. Thus, in both mammals and yeast, iron is an activator of AMPK/Snf1, whereas in mammalian cells iron overload indirectly activates the activating kinase of AMPK, in yeast iron prevents the inhibition of Snf1 by the Aft1 transcription factor. Furthermore, in bacteria, yeasts, and mammals, iron deprivation leads to an inhibition of respiratory processes.

## 4. Materials and Methods

### 4.1. Resource Availability

#### 4.1.1. Lead Contact

Further information and requests for strains and plasmids should be directed to and will be fulfilled by the lead contact Martin Kupiec (martin.kupiec@gmail.com).

#### 4.1.2. Materials Availability

All materials generated in this study are available on request to the lead contact.

#### 4.1.3. Data and Code Availability

Any additional information required to reanalyze the data reported in this paper is available from the lead contact upon request. This paper does not report original code.

### 4.2. Experimental Model and Subject Details

#### Yeasts, Plasmids, and Growth Conditions

Experiments were conducted using *S. cerevisiae*. DH5a bacteria were used for plasmid propagation and standard procedures. Strains used are listed in Table 1; plasmids used are listed in Table 2. Oligonucleotides used for mutagenesis of Snf1, Aft1, and Pma1 are listed in Table 3. All strains are related to W303a, except for PJ694, which was used for yeast 2-hybrid assays. Yeasts were transformed with DNA using the frozen lithium acetate method [81]. The construction of poly-amino acid substitutions in Snf1 and the exchange of selective markers in plasmids was by gap repair [82] and PCR.

All yeast strains are in the W303 background, except PJ694 (for yeast 2-hybrid assays).

All oligonucleotides were ordered from Sigma. Desalted, no purification. Upon receipt they were resuspended to 100 μM in 5 mM Tris (pH 8.5). Primers marked with an asterisk were phosphorylated using phosphonucleotide kinase before use.

*PMA1* (from −934 of promoter to +834 of 3′UTR) was inserted into the multiple cloning site of pRS313 and pRS314. The final 18 codons were truncated using Phusion PCR. Four or eight histidines were inserted into plasmid 847 using Phusion PCR. Primers were phosphorylated before use with polynucleotide kinase.

Standard sugar concentrations were 4% for glucose (to ensure complete repression of the *ADH2* promoter), 3% for glycerol, and 2% for ethanol, unless stated differently. For *ADH2* and *SUC2* expression assays, cells were grown in 4% glucose to ensure complete repression of expression, washed 3× with water, and resuspended in media containing 2% ethanol or 2% sucrose as indicated. Cells were grown at 30 °C. Bathophenanthrolinedisulfonate (BPS) was added at 100 μM (from 100 mM stock); Mohr’s salt (ferrous ammonium sulphate) was added at 5 mM concentration (from 0.5M stock).

### 4.3. Method Details

#### 4.3.1. β-Galactosidase Assays for Gene Expression

β-galactosidase assays were performed using log phase cells. Cells containing *prADH2* or *SUC2*::LacZ plasmids were grown overnight in 4% glucose synthetic defined medium, diluted in the morning, and grown for an additional 3 h. A sample was taken for measuring (t = 0) and cells were washed 3× with 25 mL water before resuspension in the indicated medium. After 3 h, β-galactosidase activity was measured. For *YRO2* expression experiments, cells were grown for 18 h with the indicated carbon source (and 100 μM BPS where indicated). Cell concentration was determined by reading 100 μL of cells at 595 nm. An amount of 20 μL of cells was added to the β-galactosidase reaction mix (40 μL YPER (Pierce 78990), 80 μL Z-buffer (120 mM Na_2_HPO_4_, 80 mM NaH_2_PO_4_, 20 mM KCl, 2 mM MgSO_4_), 24 μL ONPG (4 mg/mL), 0.4 μL β-mercaptoethanol) and incubated at 30 °C for 15 min. Reactions were stopped by the addition of 54 μL 1M Na_2_CO_3_. The Eppendorf tubes were centrifuged for 1 min at full speed to pellet the cell debris, and 200 μL supernatant was removed, and absorbance was read at 415 nm using a microplate reader. Miller Units were calculated by the equation Miller Units = (1000 × A_415_)/(time × volume of cells × A_595_ − 0.055, where the A_415_ and A_595_ has been corrected for blanking and path length (final path length = 1 cm). The expression was calculated as a rate of Miller Unit increase per hour and normalized as a percentage of the expression rate in WT cells (=100). For *prFET3*::LacZ experiments, this is in glucose medium with 100 μM BPS, for *prADH2*::LacZ experiments, this is in ethanol medium. Typically, in wild-type cells, *ADH2* is expressed at 1000 Miller Units per hour in ethanol medium and barely expressed in glucose medium, *SUC2* at 50 Miller Units/hour in sucrose and at 100 Miller Units/hour in ethanol medium. *FET3* is typically expressed at 20 Miller Units per hour in glucose media, increasing to 200 Miller Units per hour upon addition of 100 μM BPS. Three biological replicates were measured. Error bars are mean ±1 standard deviation. *t*-tests were performed using GraphPad *t*-test calculator. NS = not significant, * = *p* ≤ 0.05, ** = *p* ≤ 0.01, *** = *p* ≤ 0.001.

#### 4.3.2. Yeast 2-Hybrid Experiments

*S. cerevisiae* strain PJ694 or PJ694 *hsp30Δ* was transformed with the indicated plasmids expressing proteins or protein fragments fused to either GAD or GBD and *PMA1-Δ901* where indicated. Cells were grown overnight in indicated medium and diluted in the morning with the same medium for an additional 3 h before β-galactosidase activity was determined as above. Error bars are mean ±1 standard deviation. *t*-tests were performed using GraphPad *t*-test calculator. NS = not significant, * = *p* ≤ 0.05, ** = *p* ≤ 0.01, *** = *p* ≤ 0.001.

#### 4.3.3. Western Blots

Cells were grown overnight in 4% glucose, diluted in the morning to 20 mL, and grown for an additional three hours. Two 5 mL samples of glucose grown cells were removed—to one 5 μM BPS was added. The remaining 10 mL was centrifuged (4000 rpm, 1 min) and cells were washed 2× with 25 mL water and resuspended in 10 mL of 2% ethanol medium. This was also divided into two and one sample treated with 5 μM BPS. After 30 min, cells were harvested and protein-solubilized in sample buffer by the method developed by the Kuchin group to prevent activation of Snf1 by centrifugation [88]. Cells were boiled for 5 min before resuspension in 1× TE and treatment with 0.2M NaOH. Sample buffer volume was adjusted to give equal OD for all samples (30 μL/OD) and cells were boiled for 5 min. After running samples on 8% polyacrylamide gels, the proteins were wet-transferred to nitrocellulose membranes. Antibodies used were mouse α-pgk1 (Abcam, Cambridge, UK) for a loading control, mouse α-HA (Santa Cruz, Dallas, TX, USA), and rabbit α-phospo-T172 (AMPK) (Cell Signalling Technology, Danvers MA, USA), all at 1/1000 dilution. Secondary antibodies were conjugated to HRP. Images were minimally processed using ImageJ. Snf1 abundance and phosphorylation was quantified using data from Figure 1C and Figure 6A,E. Both Snf1 abundance and T210 phosphorylation was normalized to WT cells in Ethanol (set W303a00%). *t*-tests were performed using GraphPad *t*-test calculator. NS = not significant, * = *p* ≤ 0.05, ** = *p* ≤ 0.01, *** = *p* ≤ 0.001.

#### 4.3.4. Microscopy

An amount of 5 μL of log phase cells was imaged using an EVOS microscope (60× objective) with the GFP filter for GFP, YFP filter for Venus/YFP, and the Texas Red filter for Cherry. The dimensions of each panel correspond to 20 μm × 20 μm. Cells were not concentrated before imaging to prevent perturbations to Snf1 activity [89]. Images were processed using the brightness/contrast function of Image J, to give a black background. For statistics, over 300 cells were counted. Experiments were repeated at least three times on different days. All figures shown in the manuscript are of identical magnification, a size bar is provided in Figure 3C.

#### 4.3.5. Jen1-GFP Visualization

Cells were grown overnight in glycerol medium and Jen1-GFP expression was induced by the addition of 2% galactose for 4 h. Cells were washed 3× and resuspended in either 2% glucose or 3% glycerol + 0.1 mM BPS medium to determine whether iron chelation causes internalization of Jen-GFP in the same way as glucose. Whilst glucose can be added to galactose to repress GAL promoters (and thus prevent Jen1-GFP expression), when cells were treated with Glycerol + 0.1 mM BPS the galactose had to be washed away (indicated by an arrow) to prevent further Jen1-GFP expression. *Rod1Δ* and *reg1Δ* are included as known controls that do not internalize Jen-GFP in glucose media.

## Figures and Tables

**Figure 1 ijms-24-01368-f001:**
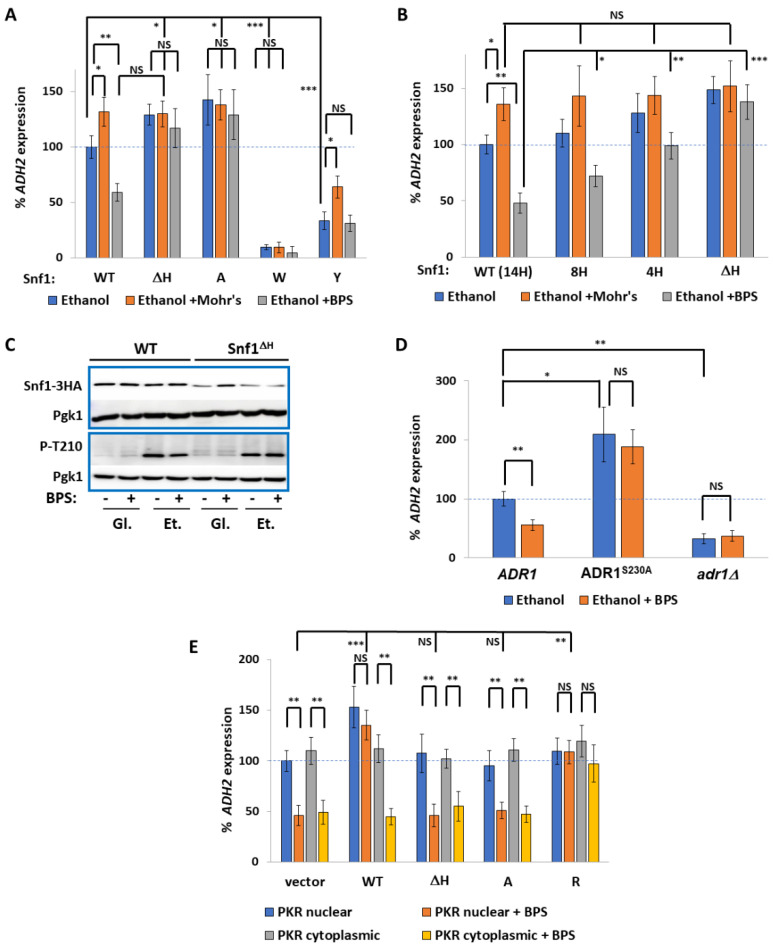
Snf1 is inhibited by iron chelation. (**A**,**B**) Rate of *prADH2*::LacZ expression in *snf1*Δ cells expressing *prSNF1::*Snf1-GFP from plasmids with the indicated mutations (deletion of polyHIS (ΔH), substitution to poly^A^ (A), substitution to poly^W^ (W), substitution to poly^Y^ (Y), four histidines (4H), eight histidines (8H)). Snf1^WT^ has 14 histidines in its polyHIS tract. An amount of 0.1 mM BPS or 5 mM Mohr’s salt was added as indicated. N = 3. Error bars are mean ±1 standard deviation. *t*-tests were performed as indicated. NS = not significant, * = *p* ≤ 0.05, ** = *p* ≤ 0.01, *** = *p* ≤ 0.001. (**C**) Western blot showing phosphorylation of Snf1 under indicated conditions. Gl.: glucose; Et.: ethanol. Snf1 abundance and phosphorylation at T210 are quantified in Appendix A and Appendix A, respectively. (**D**) Rate of *prADH2*::LacZ expression in *adr1Δ* cells expressing *ADR1* or *ADR1^S230A^* or empty vector as indicated. An amount of 0.1 mM BPS was added as indicated. N = 3. Error bars are mean ±1 standard deviation. *t*-tests were performed as indicated. NS = not significant, * = *p* ≤ 0.05, ** = *p* ≤ 0.01, *** = *p* ≤ 0.001. (**E**) Rate of *prADH2*::LacZ expression in WT W303a cells overexpressing the PKR of Snf1 (aa1-53) with indicated mutations (deletion of polyHIS (ΔH), substitution to poly^A^ (A), substitution to poly^R^ (R)) or empty vector. The PKR was fused to the Gal Binding Domain (GBD) to send it to the nucleus, or expressed without the GBD to keep it cytoplasmic. N = 3. Error bars are mean ±1 standard deviation. *t*-tests were performed as indicated. NS = not significant, * = *p* ≤ 0.05, ** = *p* ≤ 0.01, *** = *p* ≤ 0.001.

**Figure 2 ijms-24-01368-f002:**
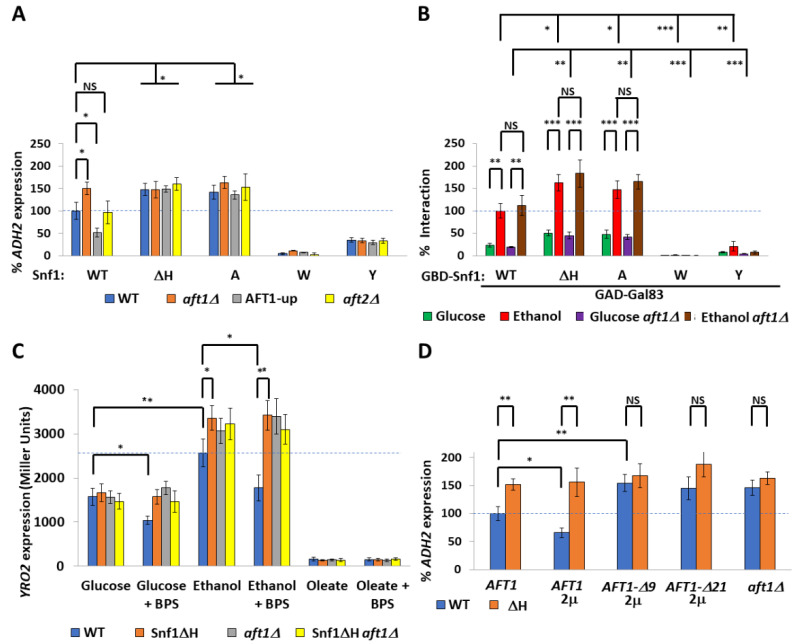
Aft1 inhibits Snf1. (**A**) Rate of *prADH2*::LacZ expression of indicated cells (in a *snf1Δ* background) cells expressing *prSNF1::*Snf1-GFP from plasmids with the indicated mutations (deletion of polyHIS (ΔH), substitution to poly^A^ (A), substitution to poly^W^ (W), substitution to poly^Y^ (Y)). An amount of 100 μM BPS or 5 mM Mohr’s salt was added as indicated. N = 3. Error bars are mean ±1 standard deviation. *t*-tests were performed as indicated. NS = not significant, * = *p* ≤ 0.05, ** = *p* ≤ 0.01, *** = *p* ≤ 0.001. (**B**) Yeast two-hybrid experiment showing interaction between GAD-Gal83 and GBD-Snf1 (with indicated mutations (deletion of polyHIS (ΔH), substitution to poly^A^ (A) substitution to poly^W^ (W), substitution to poly^Y^ (Y))) in either wild-type or *aft1Δ* cells, where 100% interaction is defined as GBD-Snf1 interacting with GAD-Gal83 in ethanol media. N = 3. Error bars are mean ±1 standard deviation. *t*-tests were performed as indicated. NS = not significant, * = *p* ≤ 0.05, ** = *p* ≤ 0.01, *** = *p* ≤ 0.001. (**C**) *YRO2* expression in *snf1Δ* and *snf1Δaft1Δ* cells bearing the indicated *prSNF1::Snf1-GFP* plasmids. Cells were grown with the indicated carbon source and *prYRO2*::LacZ expression measured after 18 h. N = 3. Error bars are mean ±1 standard deviation. *t*-tests were performed as indicated. NS = not significant, * = *p* ≤ 0.05, ** = *p* ≤ 0.01, *** = *p* ≤ 0.001. (**D**) Rate of *prADH2*::LacZ expression in *snf1Δ aft1Δ* cells bearing the indicated *prSNF1::Snf1-GFP* plasmids and the indicated Aft1 plasmids (or empty vector) grown in ethanol medium. N = 3. Error bars are mean ±1 standard deviation. *t*-tests were performed as indicated. NS = not significant, * = *p* ≤ 0.05, ** = *p* ≤ 0.01, *** = *p* ≤ 0.001.

**Figure 3 ijms-24-01368-f003:**
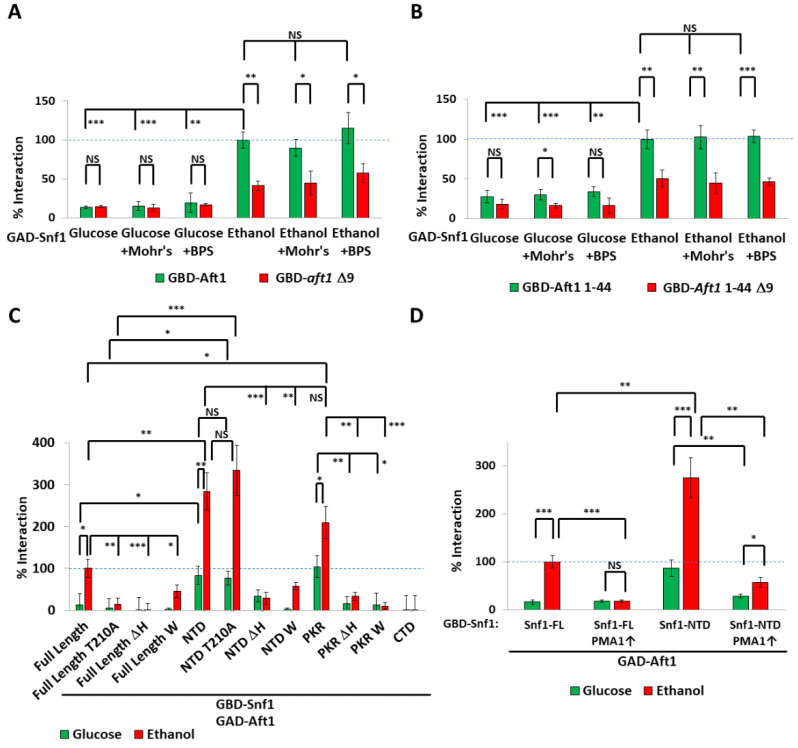
Interactions between Aft1 and Snf1 (**A**,**B**). Yeast two-hybrid experiment showing interaction between GAD-Snf1 and GBD-Aft1 (**A**) or GBD-Aft1^1-44^ (**B**) under the indicated conditions. An amount of 100 μM BPS or 5 mM Mohr’s salt was added as indicated, and 100% interaction is defined as GAD-Snf1 with GBD-Aft1 (**A**) or GBD-Aft1^1-44^ (**B**). N = 3. Error bars are mean ±1 standard deviation. *t*-tests were performed as indicated. NS = not significant, * = *p* ≤ 0.05, ** = *p* ≤ 0.01, *** = *p* ≤ 0.001. (**C**) Yeast two-hybrid experiment showing interaction between GAD-Aft1 and GBD-Snf1 fragments as described in Appendix A, and 100% interaction is defined as GAD-Aft1 with GBD-Snf1-FL in ethanol. N = 3. Error bars are mean ±1 standard deviation. *t*-tests were performed as indicated. NS = not significant, * = *p* ≤ 0.05, ** = *p* ≤ 0.01, *** = *p* ≤ 0.001. (**D**) Yeast two-hybrid experiment showing interaction between GAD-Aft1 and GBD-Snf1. FL = full length Snf1 (aa 1-633). NTD = aa1-391. Pma1 was hyperactivated (Pma1↑) by simultaneously truncating the final 18 amino acids of Pma1 (Pma1-Δ901) and deleting *HSP30* [42]. 100% interaction is defined as GAD-Aft1 with GBD-Snf1-FL in ethanol. N = 3. Error bars are mean ±1 standard deviation. *t*-tests were performed as indicated. NS = not significant, * = *p* ≤ 0.05, ** = *p* ≤ 0.01, *** = *p* ≤ 0.001.

**Figure 4 ijms-24-01368-f004:**
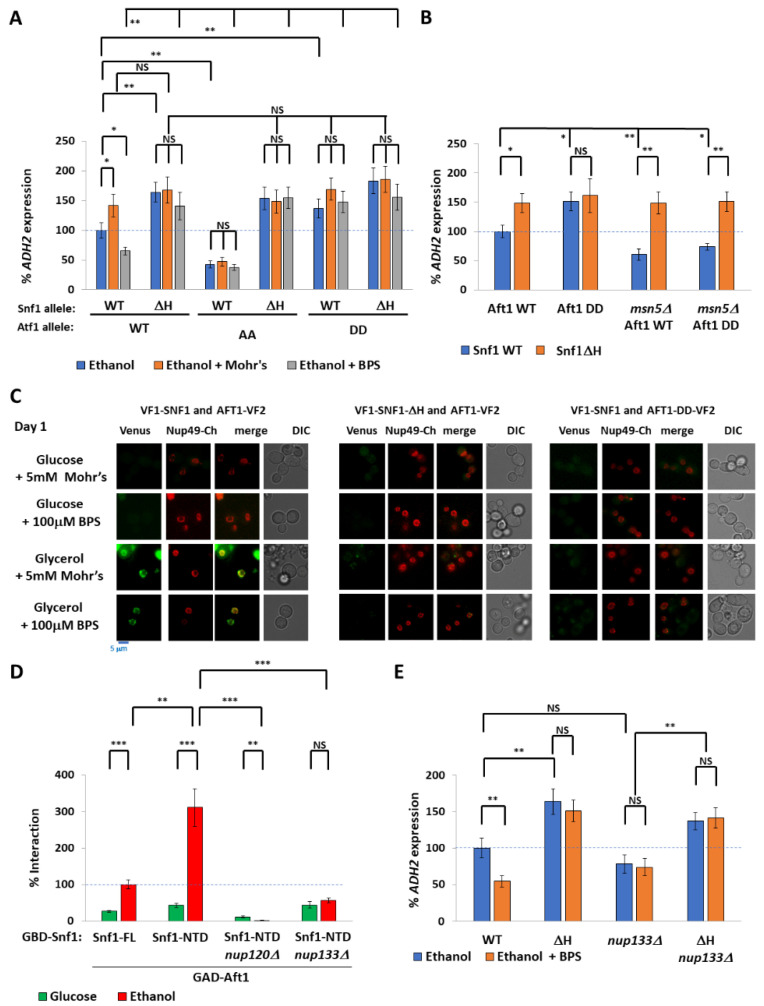
Nuclear Aft1 inhibits Snf1. (**A**,**B**) Rate of *prADH2*::LacZ expression of *snf1Δaft1Δ* cells (and *snf1Δaft1Δmsn5Δ* cells (**B**)) expressing indicated Snf1 and Aft1 mutants from plasmids. Aft1^AA^ is Aft1 –S210A S224A. Aft1^DD^ is Aft1-S210D S224D. An amount of 100 μM BPS or 5 mM Mohr’s salt was added as indicated. N = 3. Error bars are mean ±1 standard deviation. *t*-tests were performed as indicated. NS = not significant, * = *p* ≤ 0.05, ** = *p* ≤ 0.01, *** = *p* ≤ 0.001. (**C**) Protein complementation assay by split Venus to show interaction between VF1-Snf1 and Aft1-VF2. Cells were grown under indicated conditions overnight, with dilution before imaging two hours later. Aft1-DD is Aft1-S210D S224D. Nup49-Cherry was used to mark the nuclear membrane. (**D**) Yeast two-hybrid experiment to show interaction between GAD-Aft1 and GBD-Snf1 in the absence of either *nup120* or *nup133*. FL = full length Snf1 aa1-633. NTD = Snf1^1-391^. The 100% interaction is defined as GAD-Aft1 with GBD-Snf1-FL in ethanol. N = 3. Error bars are mean ±1 standard deviation. *t*-tests were performed as indicated. NS = not significant, * = *p* ≤ 0.05, ** = *p* ≤ 0.01, *** = *p* ≤ 0.001. (**E**) Rate of *prADH2*::LacZ expression of *snf1Δ* and *snf1Δnup133Δ* cells expressing either Snf1^WT^ or Snf1^ΔH^ from plasmids. An amount of 0.1 mM BPS was added as indicated. N = 3. Error bars are mean ±1 standard deviation. *t*-tests were performed as indicated. NS = not significant, * = *p* ≤ 0.05, ** = *p* ≤ 0.01, *** = *p* ≤ 0.001.

**Figure 5 ijms-24-01368-f005:**
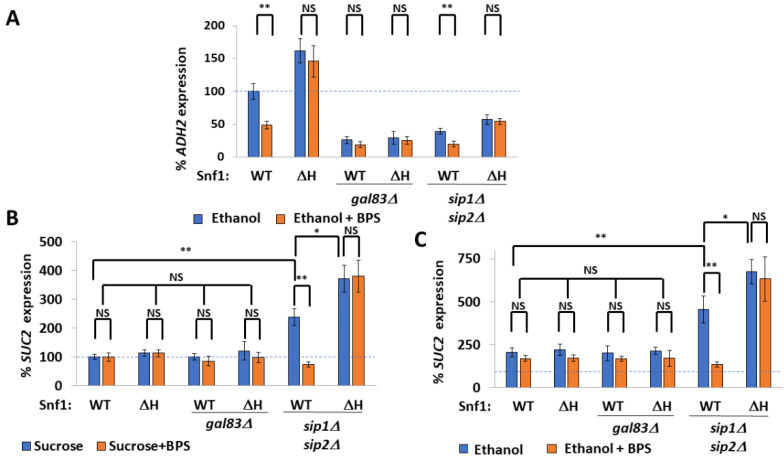
Temporal–spatial regulation of Snf1 by iron. (**A**) Rate of *prADH2*::LacZ expression when Snf1 is restricted to the cytoplasm (*gal83Δ*) or to the nucleus (*sip1Δsip2Δ*) upon iron chelation by 100 μM BPS. N = 3. Error bars are mean ±1 standard deviation. *t*-tests were performed as indicated. NS = not significant, * = *p* ≤ 0.05, ** = *p* ≤ 0.01. (**B**,**C**) Rate of *prSUC2*::LacZ expression when Snf1 is restricted to the cytoplasm (*gal83Δ*) or to the nucleus (*sip1Δsip2Δ*) upon iron chelation by 100 μM BPS: (**B**) is with sucrose as the carbon source, (**C**) is with ethanol (and is normalized to WT sucrose. N = 3. Error bars are mean ±1 standard deviation. *t*-tests were performed as indicated. NS = not significant, * = *p* ≤ 0.05, ** = *p* ≤ 0.01.

**Figure 6 ijms-24-01368-f006:**
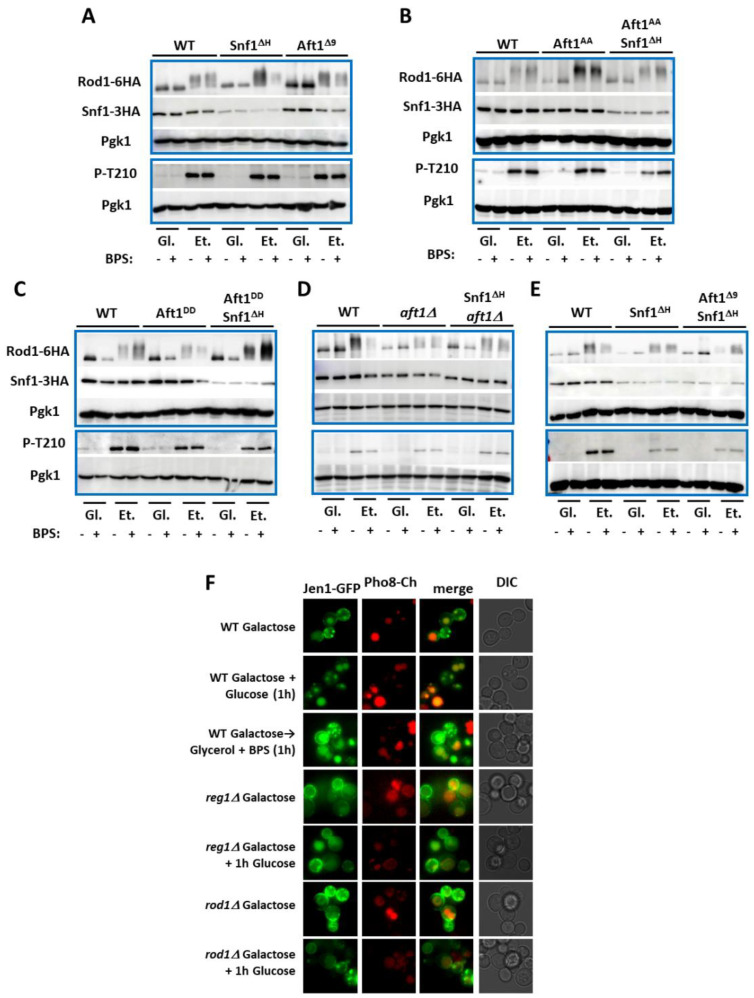
Non-nuclear substrates of Snf1 are not regulated by iron deprivation. (**A**–**E**) Western blots showing phosphorylation of Rod1-6HA, Snf1-3HA protein levels, and phosphorylation of Snf1 at T210 following transfer of indicated cells from glucose (Gl) media to ethanol (Et) media (and adding 0.1 mM BPS as indicated) for 30 min. (**F**) Internalization of Jen1-GFP. Jen1-GFP was expressed from a *GAL* promoter overnight. Either glucose at 2% was added to internalize Jen1-GFP (vacuoles are marked with Pho8-Cherry) or the galactose was washed out and replaced by glycerol + 100 μM BPS to determine if iron deprivation leads to Jen1-GFP internalization.

**Figure 7 ijms-24-01368-f007:**
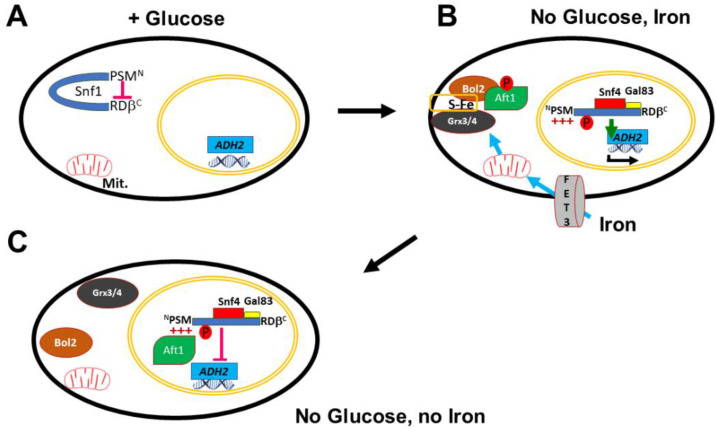
Model. (**A**) In the presence of glucose, the PSM is deprotonated and interacts with RD-γ, T210 is not phosphorylated and Snf1 is inactive. (**B**) In the absence of glucose, T210 is phosphorylated and the PSM is protonated and disengages from RD-γ. Snf4 and a γ-subunit interact with the RD. Snf1/Snf4/Gal83 localize to the nucleus and genes, such as *ADH2,* are expressed. Iron is imported and iron–sulfur clusters are manufactured in the mitochondria. These interact with Grx3/4 and Bol2 to sequester Aft1 in the cytoplasm. Aft1 is also phosphorylated by Hog1 at S210 and S224 and this leads to its export from the nucleus by Msn5. (**C**) In the absence of iron or glucose, Aft1 enters the nucleus and interacts at the nuclear membrane with the protonated PSM of Snf1, thereby inhibiting Snf1.

**Table 1 ijms-24-01368-t001:** Yeast Strains.

Number	Name	Genotype	From
1	WT	W303 MATa *ade2-1 his3-11,15 trp1-1 leu2-3,112 ura3-1 rad5-535 bud4*	Lab collection
2	*snf1*	W303a *snf1::HYG*	E. Young [25]
3	*r*eg1	W303a *MATα reg1::NAT*	E. Young [25]
4	*snf1 reg1*	W303a *snf1:: HYG reg1::NAT*	E. Young [25]
5	*snf1 h*sp30	W303a *snf1:: HYG hsp30::KAN*	[42]
6	PJ694	*trp1-901 leu2-3,112, ura3-52, his3Δ200, gal4Δ, gal80Δ, prGAL2:ADE2 lys2::prGAL1::HIS3 met2::prGAL7-LacZ*	Lab collection
7	*adr1*	W303a *adr1::NAT*	E. Young [25]
11	*snf1 aft1*	W303a *snf1::HYG aft1::NAT*	This study
12	*snf1 aft1* Rod1-6HA	W303a *snf1::HYG Rod1-6HA::NAT aft1::KAN*	This study
13	*snf1 aft2*	W303a *snf1::HYG aft2::NAT*	This study
14	*snf1 aft1 msn5*	W303a *snf1::HYG aft1::KAN msn5::NAT*	This study
15	*snf1 msn5*	W303a *snf1::HYG msn5::NAT*	This study
16	*snf1 sip1 sip2*	W303a *snf1::HYG sip1::NAT sip2::KAN*	This study
17	*gal83*	W303a *gal83::NAT*	This study
18	*snf1 gal83*	W303a *snf1::HYG gal83::NAT*	This study
19	*rod1*	W303a *rod1::NAT*	This study
20	*snf1 nup133*	W303a *snf1::HYG nup133::NAT*	This study
21	PJ694 *hsp30*	PJ694 *hsp30::NAT*	[42]
22	PJ694 *nup120*	PJ694*nup120::NAT*	This study
23	PJ694 *nup133*	PJ694*nup133::NAT*	This study

**Table 2 ijms-24-01368-t002:** Plasmids.

Number	Genotype	Backbone	Source/Notes
152	Snf1-GFP	pPRS313	M. Carlson [36]
728	Snf1-GFP	pRS315	Marker switch of 152
847	Snf1-ΔH-GFP	pRS315	polyHIS of 728 deleted
742	Snf1-A-GFP	pRS315	Alanine substituted for histidine in PKR of 728
745	Snf1-W -GFP	pRS315	Tryptophan substituted for histidine in PKR of 728
731	Snf1-Y-GFP	pRS315	Tyrosine substituted for histidine in PKR of 728
1244	Snf1-4H -GFP	pRS315	Four histidines inserted into 847
1248	Snf1-8H-GFP	pRS315	Eight histidines inserted into 847
816	Snf1-3HA	pRS313	M. Carlson [83]
819	Snf1-ΔH-3HA	pRS313	polyHIS of 816 deleted
1 or 919	*prADH2::LacZ*	pRS316	E. Young [25]
63	*prADH2::LacZ*	pRS313	Marker switch of 1
164	*prADH2::LacZ*	pRS315	Marker switch of 1
526	Nup49-Cherry	pRS314	M. Lisby [84]
589	Nup49-Cherry	pRS316	Marker switch of 526
590	Nup49-Cherry	pRS315	Marker switch of 526
1020	Adr1-3HA	2μ TRP1 KanMX	E. Young [25]
1021	Adr1^S230A^-3HA	2μ TRP1 KanMX	E. Young [25]
1081	Aft1-1up	pRS416	D. Winge [16]
1160	Aft1-1up	pRS413	Marker switch of 1081
935	*prFET3::LacZ*	Yep354 (2μ URA3)	V. Costa [50]
931	Aft1-HA	pRS416	V. Costa [50]
932	Aft1-S210D S224D-HA	pRS416	V. Costa [50]
933	GFP-Aft1	pRS426	V. Costa
934	4t-2GST-Aft1-S210A S224A	pGEX	V. Costa [50]
1208	Aft1-S210A S224A-HA	pRS416	Fragment from 934 replacing fragment in 931
1206	Aft1-HA	pRSs415	Marker switch of 931
1202	Aft1-S210D S224D-HA	pRS415	Marker switch of 932
1174	GFP-Aft1	pRS425	Marker switch of 933
1212	Aft1-S210A S224-HA	pRS415	Marker switch of 1208
1172	GFP-Aft1 Δ9	pRS426	Aa16-24 deleted in 933
1173	GFP-Aft1 Δ9	pRS425	Marker switch of 1172
1195	GFP-Aft1 Δ24	pRS426	Aa16-36 deleted in 933
1199	GFP-Aft1 Δ24	pRS425	Marker switch of 1195
1204	GFP-Aft1 S210A, S224A	pRS426	S210A S224A mutations in 933
1205	GFP-Aft1 S210D, S224D	pRS426	S210D S224D mutations in 933
1234	Pma1	pRS314	Pma1 inserted into prs314
1238	Pma1-Δ901	pRS314	Final 18aa of Pma1 deleted
216	*prSUC2::LacZ*	Yep354 (2μ URA3)	S. Kuchin [85]
722	*prYRO2*::LacZ	pRS416	Z. Liu [48]
771	*prGAL1::*Jen1-GFP	pRS316	S. Leon [57]
230	*prVph1::*Cherry-Pho8	pRS315	T. Stevens [86]
**Yeast 2-hybrid plasmids**			
153	*prADH1::GAD*	LEU2, 2μ	pACT2 yeast 2-hybrid empty vector
154	*prADH1::GBD*	TRP1, 2μ	pGBT9 yeast 2-hybrid empty vector
155	*prADH1::GBD*	URA3, 2μ	pGBU9 yeast 2-hybrid empty vector
757	GAD-Gal83	LEU2, 2μ	Gal83 in 153
762	GBD-Snf1	URA3, 2μ	Snf1 in 155
764	GBD-Snf1 T210A	URA3, 2μ	Snf1 T210A in 155
789	GBD-Snf1 ΔH	URA3, 2μ	polyHIS deletion of 762
780	GBD-Snf1 W	URA3, 2μ	Tryptophan substituted for polyHIS in 762
781	GBD-Snf1 1-391	URA3, 2μ	Snf1 1-391 in 155
849	GBD-Snf1 1-391 T210A	URA3, 2μ	Snf1 1-391 T210A in 155
795	GBD-Snf1 1-391 ΔH	URA3, 2μ	Snf1 1-391 ΔH in 155
785	GBD-Snf1 1-391 W	URA3, 2μ	Snf1 1-391 W in 155
828	GBD-Snf1 1-53	URA3, 2μ	Snf1 1-53 in 155
836	GBD-Snf1 1-53 ΔH	URA3, 2μ	Snf1 1-53 ΔH in 155
829	GBD-Snf1 1-53 A	URA3, 2μ	Snf1 1-53 A in 155
831	GBD-Snf1 1-53 R	URA3, 2μ	Snf1 1-53 R in 155
837	GBD-Snf1 1-53 W	URA3, 2μ	Snf1 1-53 W in 155
786	GBD-Snf1 392-633	URA3, 2μ	Snf1 392-633 in 155
991	GBD-Snf1 1-53	TRP1, 2μ	Snf1 1-53 in 154
993	GBD-Snf1 1-53 ΔH	TRP1, 2μ	Snf1 1-53 ΔH in 154
1013	GBD-Snf1 1-53 A	TRP1, 2μ	Snf1 1-53 A in 154
1261	GBD-Snf1 1-53 R	TRP1, 2μ	Snf1 1-53 R in 154
1012	*prADH1::Snf1* 1-53	TRP1, 2μ	Snf1 1-53 in 154 (no GBD)
1004	*prADH1::Snf1* 1-53 ΔH	TRP1, 2μ	Snf1 1-53 ΔH in 154 (no GBD)
1019	*prADH1::Snf1* 1-53 A	TRP1, 2μ	Snf1 1-53 A in 154 (no GBD)
1267	*prADH1::Snf1* 1-53 R	TRP1, 2μ	Snf1 1-53 R in 154 (no GBD)
857	GAD-Snf1	LEU2, 2μ	Snf1 1-633 in 153
1210	GBD-Aft1	TRP1, 2μ	Aft1 in 154
1217	GBD-Aft1 1-44	TRP1, 2μ	Aft1 1-44 in 154
1211	GBD-Aft1 Δ9	TRP1, 2μ	Aft1 Δ9 in 154
1218	GBD-Aft1 1-44 Δ9	TRP1, 2μ	Aft1 1-44 Δ9 in 154
1229	GAD-Aft1	LEU2, 2μ	Aft1 in 153
**Venus Constructs**			
4	*prADH1::*VF1	pRS413	P. Chartrand [87]
8	*prADH1::*VF2	pRS415	P. Chartrand [87]
1168	*prSNF1*::VF1-Snf1-3HA	pRS413	VF1 inserted at N-terminus of 816
1170	*prSNF1*::VF1-Snf1 ΔH-3HA	pRS413	VF1 inserted at N-terminus of 819
1149	*prADH1::*Aft1-VF2	pRS415	Aft1 inserted into 8
1150	*prADH1::*Aft1 S210D S224D-VF2	pRS415	Aft1 S210D S224D inserted into 8

**Table 3 ijms-24-01368-t003:** Oligonucleotides for Mutagenesis of Snf1.

Name	Description	Sequence
**Gap Repair Mutagenesis of polyHIS**		
C54	Y-5	CTACTATTATTACTATTATTACTACTATTACTATTATGGATATggcggaagcaactcgacg
C53	Y-3	cataatagtaatagtagtaataatagtaataatagtagtaGCTAGAATTTGCATTGGCAGGTG
C57	A-5	GCAGCTGCCGCAGCAGCCGCTGCCGCAGCAGCTGCAGCAGGAGCCggcggaagcaactcgacg
C58	A-3	ggctcctgctgcagctgctgcggcagcggctgctgcggcagctgcGCTAGAATTTGCATTGGCAGGTG
C74	F-5	CTTTTTCTTCTTCTTTTTCTTCTTCTTCTTCTTCTTTGGATTTggcggaagcaactcgacg
C73	F-3	caaagaagaagaagaagaagaaaaagaagaagaaaaagaaGCTAGAATTTGCATTGGCAGGTG
C107	K-5	aaaaagaaaaagaaaaagaagaaaaaaaagaaaaaaaaaggtaagGGCGGAAGCAACTCGACG
C106	K-3	cttacctttttttttcttttttttcttctttttctttttctttttGCTAGAATTTGCATTGGCAGGTG
C115	R-5	cgtagaaggagacgtcgtaggagaagacgtaggcgtaggggtaggGGCGGAAGCAACTCGACG
C114	R-3	cctacccctacgcctacgtcttctcctacgacgtctccttctacgGCTAGAATTTGCATTGGCAGGTG
C104	W-5	tggtggtggtggtggtggtggtggtggtggtggtggtggggttggGGCGGAAGCAACTCGACG
C103	W-3	ccaaccccaccaccaccaccaccaccaccaccaccaccaccaccaGCTAGAATTTGCATTGGCAGGTG
C117	Delta-5	GGCGGAAGCAACTCGACG
C116	Delta-3	caccatccgctaaggacgacttgggattgtttagcgtcgagttgcttccgccGCTAGAATTTGCATTGGCAG
**Mutagenesis of Aft1**		
C271	I15-rev	AATCGGTGACGCATGTTCTATG
C272	Δ9-fwd	GAAGGCTTCAATCCGGCTGACATAGAACATGCGTCACCGATTTTTGTATATGCTCTACCCAAAAGTGC
C275	Δ21-fwd	GTAGTCAACCATAATGAGGGTCG
**Mutagenesis of Snf1**		
M69 *	4H-3	ATGGTGATGGTGGCTAGAATTTGCATTGGCAGGTGCTGTG
M70 *	8H-3	ATGGTGGTGATGGTGGTGGTGGTGGCTAGAATTTGCATTGGCAGGTGCTGTG
**Pma1 Truncation**		
C292	PMA1-5	gaattgtaatacgactcactatagggcgaattggagctccgcttcctgaaacggagaaac
C293	PMA1-3	cctcactcattaggcaccccaggctttacactttatgcttccggctccgtaaaggtatttcgcggagg
C294 *	Pma1-trunc5	TAAtcctgttgaagtagcatttaatc
C295 *	Pma1-trunc3	GACACTTCTGGTAGACTTCTTTTC

## Data Availability

Not applicable.

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
