# Peer review of "The polyHIS Tract of Yeast AMPK Coordinates Carbon Metabolism with Iron Availability"

_ijms, 2023, doi:10.3390/ijms24021368_

Round 1
Reviewer 1 Report
In this manuscript, the authors show that yeast Snf1/AMPK is inhibited under conditions of iron deficiency, due to interaction of its poly-histidine tract with Aft1, a low iron sensing transcription factor. The inhibition occurs specifically in the nucleus, to restrict adaptive responses related to iron-dependent processes such as mitochondrial respiration. The results are interesting and innovative but the text is often confusing, there are several concerns that need to be addressed and the manuscript has to be extensively revised.
Major points:
1. The title is vague and should be altered to reflect the findings of this manuscript.
2. Graphs in figures: a statistical analysis should be provided.
3. Fig. 1: the legend should explain what is W and Y in 1A or R in 1E. Is WT and WT (14H) the same? “in plasmid 154” should be deleted; in 1C, Snf1 phosphorylation (mean and SD) should be quantified.
4. Fig. S2B-C: authors must show the expression without BPS.
5. Images in Fig. 4C and S3B are rather small. Please provide imagens with a better resolution. Plus, Snf1-Aft1 interaction was observed in glycerol grown cells, both under iron supplementation and iron depletion and the images suggest that iron supplementation increased the interaction. These results do not seem to be consistent with the proposed model. In the discussion, the authors mention that “Growing cells ab initio in glycerol results in interaction of Snf1 and Aft1 and inhibition of Aft1” but they do not show the analysis of Snf1-Aft1 interaction in cells grown in glycerol (without iron supplementation or deprivation) or any data evaluating Aft1 activity.
6. The authors show that protonation of the polyHIS enables in the interaction of Snf1 with Aft1 and hyper-activation of Pma1 lowers the Snf1/Aft1 interaction. Are Pma1 and intracellular acidification affected by iron depletion?
7. Lines 337-339: “ADH2 expression is decreased in sip1sip2 cells, revealing a role for cytoplasmic Snf1 in Adr1 regulation; ADH2 expression in these cells is reduced by iron-deprivation and this is suppressed by Snf1 deltaH”. Since sip1sip2 cells are enriched in nuclear Snf1, the authors should explain why ADH2 expression decreases in this strain. The suggestion that cytoplasmic Snf1 plays a role in Adr1 regulation is not convincing to me, in particular taking into account that Snf1-polyHis seems to interact with Aft1 in the nucleus.
8. Lines 345-347, “Thus, regulation by iron-depletion and by the polyHIS tract correlates with a requirement for nuclear Snf1.”: this conclusion cannot be taken based on the results presented before this sentence. It seems to me that it can be deleted.
9. Lines 363-365 and Fig. 6A: the analysis of Dga1-GFP cannot be used as a readout of lipid droplets. For that, the authors can label the cells with BODIPY493/503 and measure fluorescence intensity by flow cytometry. However, the results show an increase of Dga1-GFP in reg1 vs wt cells (glycerol), not the decrease mentioned in text (and shown in the quantification in Fig. S4A).
10. Line 367, “iron depletion does not stimulate lipid drop production”: although the analysis of Dga1-GFP cannot be used to assess lipid droplets, the results show an increase of Dga1-GFP with iron depletion, which seems inconsistent with the proposal that non-nuclear substrates of Snf1 are not regulated by iron deprivation.
11. The discussion should be more focused. Section 3.4 is particularly confusing. Lines 510-513 are speculative since this study did not assess CCC1 expression.
Minor points:
Lines 27-28, “elimination of peroxide”: to be clearer, I suggest “elimination of hydrogen peroxide”
Line 35, “induces expression of proteins”: it should be “induces expression of genes”
Line 36, “sequester excess iron-sulphur complexes”: it should be “sequester excess iron”
Line 46, “mainly involved in”: I suggest “mainly associated with” (mRNA are translated, not involved in indicated processes)
Line 53: authors refer to Bol2 (also in lines 150-151) but Aft TF are regulated by both Fra1 and Fra2 (=Bol2). Please correct.
Line 79, authors probably mean “leads to Snf1 SUMOylation”
Line 83, “(RD- )”: please correct (throughout the manuscript, including supplementary figures).
Line 88, “domain - in S. cerevisiae the PKR is amino acids 1-53”: Please correct; I suggest “domain. In S. cerevisiae the PKR is formed by amino acids 1-53”.
Line 91: delete “,” after “Pma1”.
Results should be written in the past tense.
Numerous symbols (really high number) are not shown (delta (e.g., line 105, should be deltaH), micro (e.g., line 125), …; maybe a problem in the conversion to PDF?). The manuscript has to be extensively revised.
Lines 112-113, “the response to iron deprivation inversely correlates with histidine number”: the authors should improve the sentence, indicating what was measured (ADH2 expression).
Line 149: “manufacture” sounds weird. I suggest “production”.
Line 155: “Aft1 also possesses extended acidic, basic and polyQ regions (Figure S2A)”. It should be highlighted in Fig. S2A and mentioned in its legend.
Line 156: “of WT” should be “in WT”.
Line 164: “reduced” should be “oxidized”.
Line 174: “coA” should be “CoA”.
Fig. 2: the growth conditions in Fig. 2D must be indicated.
Line 231: “Snf1NTD” or “Snf1PKR”?
Line 250: “F” should be “B”.
Line 252: “Figure S1A”?
Line 256: hsp30” should be “HSP30” (also in line 237). “/” should be “.”.
Line 278: “of Snf1WT” should be “in Snf1WT”
Lines 353and 355: the figure does not show a rate.
Line 345: delete “.” In “expression.”
Line 355: delete “ADH2 expression”
Fig. S4A legend: “4D” should be “4A”
Fig. S4B: legend must indicate what is being quantified. Is it “% cells with Jen1-GFP”? Authors probably mean “% cell with Jen1-GFP at the plasma membrane”; “4J” should be “6G”. Notably, S4B shows the quantification in WT + BPS (glucose) but the correspondent figure is not shown in Fig. 6G.
Line 369: Rod1 is not directly involved in vacuolar degradation. It plays a role in ubiquitin-dependent endocytosis. Please correct the sentence.
Lines 406 and 437: add “.” after “expression” and ”[66]”.
Lines 444: Snf1KD?
Line 457: the meaning of the sentence is unclear. Please rephrase.
Line 488: “plasma membrane degradation”?
Line 497, “despite being 150% more active”: this is not described in results.
Line 577: delete “;”.
Lines 622-623: delete genotype (information already in table 1).
Table 1, name of strains: for yeast mutants, all letter are usually in lower case.
Table 1, genotype: is weird to refer “As 1” or “As 6”. Please replace by the name of the strain.
Table 2, genotype: Genes should be indicated in upper case and italic; vectors should be pRS…
Table 2, plasmids 1020-1021 backbone: 2 TRP1 KAN?; plasmids 935 and 216: 2 URA3?
Fig. 7 legend refers to 2A and 2B but they are not shown in the figure (only 2 is shown).
Figure legends, “Error bars are ±1 standard deviation”: please correct.
Ref. 61 seems inappropriate. It should be replaced by https://doi.org/10.1074/jbc.M116.733923
All references: the numbering is duplicated.
Reviewer 2 Report
Glucose control in yeast has been extensively studied, yet how other signals become integrated into nutritional responses is not clear. The yeast AMPK-type kinase Snf1 is a major regulator of the response to glucose limitation, by controlling the expression of genes that metabolize poor carbon sources. The authors previously showed that Snf1 has a HIS tract that can act as a pH sensor. Here, the authors explore how iron availability might also regulated by this protein. The authors show that Snf1 associates with and is inhibited by a transcription factor, Atf1. The sites of this interaction are mapped to the HIS-rich domain of Snf1. The interaction was independently validated by PCA, which led to identification of the interaction between the proteins at the nuclear membrane. Many genetic tools were developed and utilized to explore this regulatory mechanism. Although there is a lot of data in this manuscript, there are some problems in the lack of controls, clarity, and data interpretation. It appears that many of these issues could be easily addressed to make the study clearer, more accurate, and more impactful. Specifically are detailed comments outlined below.
1. Statistical tests. There are no statistical tests (p-values) reported in the paper, despite the fact that relatively small differences are reported and emphasized. For example, the authors claim in Figure 1 (and other figures) that “addition of the iron chelator BPS (0.1mM) lowers ADH2 expression by approximately 50% whilst addition of Mohr's salt (ferrous ammonium sulfate, 5mM) increases 102 ADH2 expression (Figure 1A).” The increase is from 100 to ~ 130. Is this increase statistically significant? Given that many experiments were performed in triplicate, p-values should be easily generated, especially for key findings that show subtle (less than 2-fold) differences.
2. It should be clarified that iron is not being sensed directly by the poly-histidine region, which was the overall impression I had until looking carefully at the details in the paper.
3. Consistency between what is said and what is shown. The authors state that “However, iron chelation with BPS did not affect T210 phosphorylation neither in WT cells nor in cells expressing Snf1H (Figure 1C).” This may be true, however, there are dramatic differences in phosphorylation in the figure that are not discussed and should be mentioned.
4. Figure 1E, PKR nuclear: was this confirmed to be nuclear by any test?
5. Figure 2B, how is % interaction measured for two-hybrid? By growth of cells on a plate? By a transcriptional reporter?
6. Figure 3, A and B. Why are three experiments shown for each interaction? Are these separate isolates?
7. Presumably the two hybrid interactions do not activate the reporter on their own, although this is not shown.
8. The fluorescent images are small and difficult to see.
9. Can the fluorescence intensity be quantitated?
10. Figure 5 “Rate of ADH2 expression” what test is being used? Q-RT-PCR? lacZ? What is meant by ‘rate’?
11. Figure 6. Five multi-panel blots are summarized in one paragraph that reveal some interesting findings. In particular, it appears that Snf1 lacking its HIS domain is found at lower levels in the cell. Moreover, this appears to depend on Atf1. However, the authors argue that Snf1 levels depend on the interaction between the proteins, which may not necessarily be the case. Perhaps the authors can expand on and re-visit the interpretation of this interesting data. Perhaps show the key data in the main body.
12. Can the model be simplified for clarity? Perhaps by focusing on the interaction between Snf1 and Atf1?
Round 2
Reviewer 1 Report
The manuscript was significantly improved but still needs some corrections.
The authors indicate the statistical analysis in figure legends but the “*” are not shown in the figures.
All figure legends and methods: please correct “Error bars are ±1 standard deviation” – suggestion: Error bars are mean ± standard deviation.
Figure S1 legend: since the figure is related to Fig.1, “Data from Figures 1C, 6A and 6E was used” seems to be incorrect. The quantification is related to Fig.1C.
Line 184: correct “ADH2”, it should be Adh2.
Lines 186 and 187: “coA” should be “CoA”.
Fig. 2: the growth conditions in Fig. 2D must be indicated.
Line 280: “Figure S1A” does not show what is mentioned in the legend.
Fig.6A must be deleted and the remaining (B-G) corrected (A-F).
Line 424: S4B should be S4A (or simply S4 since it is the only figure).
Line 560: delete “[10]”.
Author Response
There was a mis-understanding, and the revised paper contained the old figures. That's why there were no asterisks and the legends did not match the figures. We have now uploaded the new figures again.
We have introduced all the other changes requested. We thank the referee for the comments, which improved our paper.
Reviewer 2 Report
The figure legends in the marked copy now refer to p-values as asterisks, but I do not see any asterisks on the figures?
Author Response
There was a mis-understanding, and the revised paper contained the old figures. That's why there were no asterisks and the legends did not match the figures. We have now uploaded the new figures again.
We thank the referee for the comments, which improved our paper.